# DATA: Differentiable ArchiTecture Approximation

**Jianlong Chang**[1,2,3] **Xinbang Zhang**[1,2] **Yiwen Guo**[4,5] **Gaofeng Meng**[1]
**Shiming Xiang**[1,2] **Chunhong Pan**[1]
[1]NLPR, Institute of Automation, Chinese Academy of Sciences
[2]School of Artificial Intelligence, University of Chinese Academy of Sciences
[3]Samsung Research China - Beijing, [4]Intel Labs China, [5]Bytedance AI Lab
{jianlong.chang, xinbang.zhang, gfmeng, smxiang, chpan}@nlpr.ia.ac.cn
guoyiwen.ai@bytedance.com

## Abstract

Neural architecture search (NAS) is inherently subject to the gap of architectures during searching and validating. To bridge this gap, we develop *Differentiable ArchiTecture Approximation* (DATA) with an *Ensemble Gumbel-Softmax* (EGS) estimator to automatically approximate architectures during searching and validating in a differentiable manner. Technically, the EGS estimator consists of a group of Gumbel-Softmax estimators, which is capable of converting probability vectors to binary codes and passing gradients from binary codes to probability vectors. Benefiting from such modeling, in searching, architecture parameters and network weights in the NAS model can be jointly optimized with the standard back-propagation, yielding an end-to-end learning mechanism for searching deep models in a large enough search space. Conclusively, during validating, a high-performance architecture that approaches to the learned one during searching is readily built. Extensive experiments on a variety of popular datasets strongly evidence that our method is capable of discovering high-performance architectures for image classification, language modeling and semantic segmentation, while guaranteeing the requisite efficiency during searching.

## 1 Introduction

In the era of deep learning, how to design proper network architectures for specific problems is a crucial but challenging task. However, designing architecture with state-of-the-art performance typically requires substantial efforts from human experts. In order to eliminate such exhausting engineering, many neural architecture search (NAS) methods have been devoted to accomplishing the task automatically [14, 27, 55], *i.e.*, evolution-based NAS [13, 18, 26, 41, 43, 44, 45, 47], reinforcement learning-based NAS [2, 3, 21, 42, 56, 59, 60], and gradient-based NAS [11, 34, 35, 46, 53], which has achieved significant successes in a multitude of fields, including image classification [4, 12, 21, 30, 31, 34, 44, 53, 60], semantic segmentation [8, 32] and object detection [9, 15, 50, 52, 60].

Although the achievements in the literature are brilliant, these methods are still hard to effectively bridge the gap between architectures during searching and validating. That is, feasible paths in a learned architecture are dependent on each other and become deeply coupled during searching. In validating, however, the inherited architectures from searching always decouple the dependent paths rudely, such as DARTS [34] and SNAS [53] that choose only one path in validating. As a result, the effectiveness of the searched architectures are unclear although they could surpass the random ones.

In order to eliminate the limitation, Differentiable ArchiTecture Approximation (DATA) is proposed to elegantly minimize the gap of architectures during searching and validating. For this purpose, we develop the *Ensemble Gumbel-Softmax* (EGS) estimator, an ensemble of a group of Gumbel-Softmax

estimators, which is in a position to sample an architecture that approaches the one during searching as close as possible, while maintaining the differentiability of a promising NAS pipeline for requisite efficiency. That is, our EGS estimator suffices to not only decouple the relationship between different paths in learned architectures but also pass gradient seamlessly, yielding an end-to-end mechanism of searching deep models in a large enough search space.

To sum up, the main contributions of this work are:

- By generalizing the Gumbel-Softmax estimator, we develop the EGS estimator, which provides a successful attempt to effectively and efficiently perform structural decisions like policy gradient in the reinforcement-learning, with higher efficiency.

- With the EGS estimator, the DATA model can seamlessly bridge the gap of architectures between searching and validating, and be learned with the standard back-propagation, yielding an end-to-end mechanism of searching deep models in a large enough search space.

- Extensive experiments strongly demonstrate that our DATA model consistently outperforms current NAS models in searching high-performance convolutional and recurrent architectures for image classification, semantic segmentation, and language modeling.

## 2 Differentiable architecture search

Before introducing our approach, we first briefly review NAS. Without loss of generality, the architecture search space $\mathcal{A}$ can be naturally represented by directed acyclic graphs (DAG) each consisting of an ordered sequence of nodes. For a specific architecture, it always corresponds to a graph $\alpha \in \mathcal{A}$, represented as $\mathcal{N}(\alpha, w)$ with network weights $w$. Intrinsically, the goal in NAS is to find a graph $\alpha^* \in \mathcal{A}$ that minimizes the validation loss $\mathcal{L}_{val}(\mathcal{N}(\alpha^*, w^*))$, where the network weights $w^*$ associated with the architecture $\alpha^*$ are obtained by minimizing the training loss $w^* = \arg\min_w \mathcal{L}_{train}(\mathcal{N}(\alpha^*, w))$, *i.e.*,

$$\min_{\alpha \in \mathcal{A}} \mathcal{L}_{val}(\mathcal{N}(\alpha, w^*)), \quad s.t. \ w^* = \arg\min_w \mathcal{L}_{train}(\mathcal{N}(\alpha^*, w)). \tag{1}$$

$$\min_{\alpha \in \mathcal{A}} \mathcal{L}_{val}(\mathcal{N}(\alpha, w^*)), \quad s.t. \ w^* = \arg\min_w \mathcal{L}_{train}(\mathcal{N}(\alpha^*, w))$$

This implies that the essence of NAS is to solve a bi-level optimization problem, which is hard to optimize because of the nested relationship between architecture parameters $\alpha$ and network weights $w$. To handle this issue, we parameterize architectures with binary codes, and devote to jointly learning architectures and network weights in a differentiable way.

### 2.1 Parameterizing architectures with binary codes

For simplicity, we denote all DAGs with $n$ ordered nodes as $\mathcal{A} = \{e^{(i,j)} | 1 \leq i < j \leq n\}$, where $e^{(i,j)}$ indicates a directed edge from the $i$-th node to the $j$-th node. Corresponding to each directed edge $e^{(i,j)}$, there are a set of candidate primitive operations $\mathcal{O} = \{o_1, \cdots, o_K\}$, such as convolution, pooling, identity, and zero. With these operations, the output at the $j$-th node can be formulated as

$$x^{(j)} = \sum_{i<j} o^{(i,j)}(x^{(i)}) \tag{2}$$

where $x^{(i)}$ denotes the input from the $i$-th node, and $o^{(i,j)}(\cdot)$ is a function applied to $x^{(i)}$ which can be decomposed into a superposition of primitive operations in $\mathcal{O}$, *i.e.*,

$$o^{(i,j)}(x^{(i)}) = \sum_{k=1}^{K} A_k^{(i,j)} \cdot o_k(x^{(i)}), \quad s.t. \ A_k^{(i,j)} \in \{0,1\}, \ 1 \leq k \leq K, \tag{3}$$

where $o_k(\cdot)$ is the $k$-th candidate primitive operation in $\mathcal{O}$, and $A_k^{(i,j)}$ signifies a binary weight to indicate whether the operation $o_k(\cdot)$ is utilized on the edge $e^{(i,j)}$. For a network, by such definition, there is one and only one architecture code $\mathbf{A} \in \{0,1\}^{n \times n \times K}$ that corresponds to it, which implies that we can learn the code $\mathbf{A}$ to approximate the optimal architecture in $\mathcal{A}$.

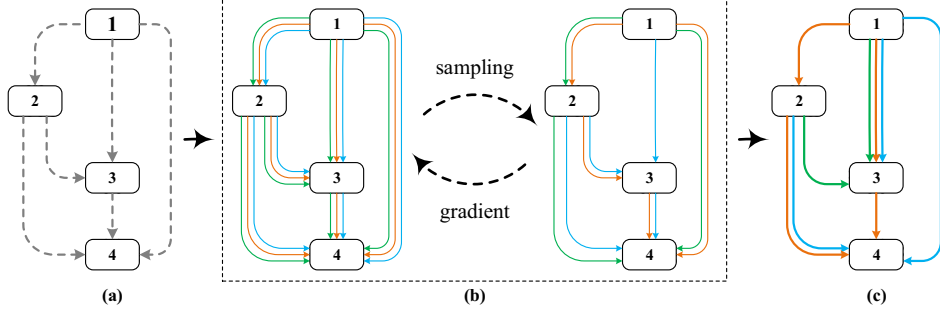

Figure 1: A conceptual visualization for the searching process within our model. (a) First, an architecture (*i.e.*, directed acyclic graph) consisting of four ordered nodes is predefined. (b) During the forward propagation, with three candidate primitive operations (*i.e.*, green, orange and cyan lines), the binary function $f(\cdot)$ is employed to generate a network in a differentiable manner. During the backward propagation, the standard back-propagation algorithm is utilized to simultaneously calculate the gradients of the both architecture parameters and network weights. (c) Finally, the details of the cell can be generated with the binary function $f(\cdot)$ and utilized to handle specific tasks.

## 2.2 From probability vectors to binary codes

Benefiting from the uniqueness property of our architecture code $\mathbf{A}$, the task of learning an architecture can therefore be converted to learning the optimal binary code $\mathbf{A}$. However, it is a fussy NP-hard problem, and is stubborn to learn directly. To overcome the obstacle, we introduce a binary function $f(\cdot)$ to approach the optimal binary codes with probability vectors, which can be easily obtained in deep models. Formally, the categorical choice in Eq. (3) can be rewritten as follows

$$\tilde{o}^{(i,j)}(x^{(i)}) = \sum_{k=1}^{K} f\left(P_k^{(i,j)}\right) \cdot o_k(x^{(i)}),$$

$$s.t. \sum_{k=1}^{K} P_k^{(i,j)} = 1, \ P_k^{(i,j)} \geq 0, \ f\left(P_k^{(i,j)}\right) \in \{0,1\}, \ 1 \leq k \leq K, \tag{4}$$

where $P_k^{(i,j)}$ is the $k$-th element in the probability vector $\mathbf{P}^{(i,j)} \in \mathbb{R}^K$ and denotes the probability of choosing the $k$-th operation on the edge $e^{(i,j)}$, and $f(\cdot)$ represents a binary function that suffices to map a probability vector to a binary code and pass gradients in a continuous manner. Specifically, $f(\cdot)$ is chosen to be a monotonically increasing function in our method, *i.e.*,

$$f(P_{k_1}^{(i,j)}) \leq f(P_{k_2}^{(i,j)}) \ \text{if} \ P_{k_1}^{(i,j)} \leq P_{k_2}^{(i,j)}, \quad 1 \leq k_1, k_2 \leq K. \tag{5}$$

By substituting $A_k^{(i,j)}$ with $f(P_k^{(i,j)})$ and considering $P_k^{(i,j)}$ instead as the variable to be optimized, we have successfully achieved a continuous relaxation. Benefiting from the flexibility of our formulation, furthermore, the optimization of NAS in Eq. (1) can be seamlessly jointed together, *i.e.*,

$$\min_{w,\alpha} \mathbb{E}_{\alpha \sim \mathbf{P}}[\mathcal{L}_{train}(\mathcal{N}(\alpha, w))], \tag{6}$$

where $\alpha \sim \mathbf{P}$ signifies that an architecture $\alpha$ and $\mathbf{P} \in \mathbb{R}^{n \times n \times K}$ are one-to-one correspondence.

With the objective in Eq. (6), the main process of optimizing it is to minimize the expected performance of architectures associated with $K$ probability vectors $\mathbf{P} \in \mathbb{R}^{n \times n \times K}$. That is, the network $\alpha$ is first generated from the binary function $f(\cdot)$ and $\mathbf{P}$. Afterward, the loss on the training dataset can be calculated by forward propagation. Relying on this loss, the gradients of the network architecture parameter $\mathbf{P}$ and the network parameter $w$ are yielded to modify these parameters better. Because of the differentiability, our model can be trained end-to-end by the standard back-propagation algorithm. In the end, the network architecture $\alpha$ is identified by $\mathbf{P}$, and the network parameter $w$ is estimated by retraining on the training set. A conceptual visualization of such a process is illustrated in Figure 1.

## 3 Architecture approximation - Ensemble Gumbel-Softmax (EGS)

Although the reformulation presented in Section 2.2 makes the search space continuous, how to define the binary function $f(\cdot)$ as desired to map each probability to a binary code needs to be sorted

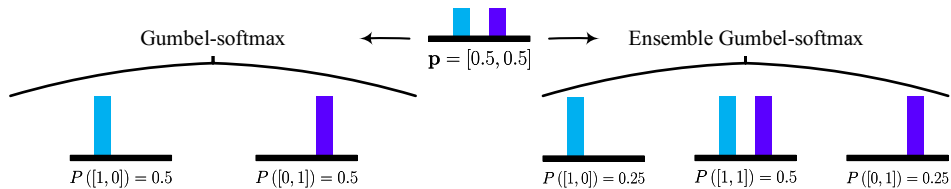

Figure 2: A visualized comparison between Gumbel-Softmax (left) and ensemble Gumbel-Softmax (right, $M = 2$). For a probability vector $\mathbf{p} = [0.5, 0.5]$, Gumbel-Softmax solely pertains to sample only two binary codes with the same probability, *i.e.*, $P([1,0]) = P([0,1]) = 0.5$. In contrast, our ensemble Gumbel-Softmax is capable of sampling more diversified binary codes, *i.e.*, $[1,0]$, $[1,1]$ and $[0,1]$. Furthermore, the probabilities of sampling these binary codes are logical. Typically, it is conceptually intuitive that the probability of sampling $[1,1]$ is larger than the probabilities of sampling the others since the probabilities in $\mathbf{p} = [0.5, 0.5]$ are equal to each other.

out. For a coarse $f(\cdot)$, it may aggravate the gap between architectures during searching and validating, such as DARTS [34] and SNAS [53] that strictly limit the binary codes as one-hot vectors. As for a refined $f(\cdot)$, we introduce an Ensemble Gumbel-Softmax (EGS) estimator to optimize the NAS problem with a principled approximation. As such, our model can be directly optimized with the back-propagation algorithm in an end-to-end way, and bridges the gap between architectures during searching and validating as close as possible, yielding an efficient and effective searching mechanism.

### 3.1 Gumbel-Softmax (GS) - to one-hot vectors

A natural formulation for representing discrete variable is to use the categorical distribution. However, partially due to the inability to back-propagate information through samples, it seems rarely applied in deep learning. In this work, we resort to the Gumbel-Max trick [17] for enabling back-propagation and and representing the process of taking decision as sampling from a categorical distribution, in order to perform NAS in a principled way. Specifically, given a probability vector $\mathbf{p} = [p_1, \cdots, p_K]$ and a discrete random variable with $P(L = k) \propto p_k$, we sample from the discrete variable $L$ by introducing the Gumbel random variables. To be more specific, we let

$$L = \arg \max_{k \in \{1, \cdots, K\}} \log p_k + G_k, \tag{7}$$

where $\{G_k\}_{k \leq K}$ is a sequence of the standard Gumbel random variables, and they are typically sampled from the Gumbel distribution $G = -\log(-\log(X))$ with $X \sim U[0, 1]$. An obstacle to directly using such approach in our problem is that the argmax operation is not really continuous. One straightforward way of dealing with this problem is to replace the argmax operation with a softmax [25, 37]. Formally, the Gumbel-Softmax (GS) estimator can be expressed as

$$\hat{L}_k = \frac{\exp\left((\log p_k + G_k)/\tau\right)}{\sum_{k=1}^{K} \exp\left((\log p_k + G_k)/\tau\right)}, \ 1 \leq k \leq K, \tag{8}$$

where $\hat{L}_k$ indicates the probability that $p_k$ is the maximal entry in $\mathbf{p}$, and $\tau$ is a temperature. When $\tau \to 0$, $[\hat{L}_1, \cdots, \hat{L}_K]$ converges to an one-hot vector, and in the other extreme it will become a discrete uniform distribution with $\tau \to +\infty$.

From the expression in Eq. (8), we see that GS estimator pertains solely to deal with the problems that only one category requires to be determined, *i.e.*, the outputs are one-hot vectors not any binary code. In NAS, however, an optimal architecture may require multiple operations on one edge, considering the practical significance [19, 49]. For instance, the residual module $\mathbf{y} = F(\mathbf{x}) + I(\mathbf{x})$ in ResNets [19] consists of two operations with a learnable mapping $F(\cdot)$ and the identity $I(\cdot)$. That is, choosing different operations in $\mathcal{O}$ may not be mutually exclusive but compatible. One direct way of handling this limitation is to map all possible operation combinations to $2^K$-dimensional vectors, where $K$ is the number of candidate operations in $\mathcal{O}$. However, it seems difficult to search architectures efficiently when there are many candidate operations, *i.e.*, $K$ is really large.

## 3.2 Ensemble Gumbel-Softmax (EGS) - to any binary code

In order to address the aforementioned limitation in the traditional GS estimator, we propose an Ensemble Gumbel-Softmax (EGS) estimator to model the binary function $f(\cdot)$ formulated in Eq. (4), which is capable of choosing diversiform numbers of operations on different edges. To this end, architectures are equally recoded into a group of one-hot vectors that can be sampled from probability vectors with the GS estimator. Because of the equivalency, in turn, any architecture is sampled by compositing the results from the GS estimator.

For clarity of exposition, the recoding of $\mathbf{A}^{(i,j)} \in \{0,1\}^K$ is described only, where $\mathbf{A}^{(i,j)}$ implies the chosen operations on an edge $e^{(i,j)}$. Naturally, such a $K$-dimensional vector $\mathbf{A}^{(i,j)} \in \{0,1\}^K$ can be recoded into a superposition of $K$ one-hot vectors, $i.e.$,

$$\mathbf{A}^{(i,j)} = \sum_{k=1}^{K} v_k \cdot \mathbf{a}_k^{(i,j)}, \quad v_k \in \{0,1\}, \ 1 \le k \le K, \tag{9}$$

where $\mathbf{a}_k^{(i,j)} \in \mathbb{R}^K$ is a $K$-dimensional one-hot vector that uniquely corresponds to the operation $o_k \in \mathcal{O}$, $v_k = 1$ implies that the operation $o_k$ is chosen on edge $e^{(i,j)}$, and $v_k = 0$ otherwise. Benefiting from the equivalency, any architecture code can be represented with a group of one-hot vectors, and one-hot vectors also can be sampled from probability vectors with the GS estimator. Intrinsically, such straightforward process can be considered as the inverse operation of the binary function $f(\cdot)$. That is, the problem of modeling the binary function $f(\cdot)$ can be recast as to find the inversion of such process.

Inspired from the above relationship between architecture codes, one-hot vectors and probability vectors, we model the binary function $f(\cdot)$ by introducing the inversion of this relationship. In Figure 2, a visualized comparison between the GS and EGS estimators intuitively shows that our EGS estimator is more excellent than the GS estimator, in terms of both sampling capability and rationality in practice. Given a probability vector, the EGS estimator, an ensemble of multiple GS estimators is profound for sampling any binary code, $i.e.$,

**Definition 1.** *For a $K$-dimensional probability vector $\mathbf{p} = [p_1, \cdots, p_K] \in \mathbb{R}^K$ and $M$ one-hot vectors $\{\mathbf{z}^{(1)}, \cdots, \mathbf{z}^{(M)}\}$ sampled from $\mathbf{p}$ with the GS estimator, the $K$-dimensional binary code $\mathbf{b} = [b_1, \cdots, b_K] \in \{0,1\}^K$ sampled with EGS is*

$$b_k = \max_{1 \le i \le M} \left( z_k^{(i)} \right), 1 \le k \le K$$

*where $M$ is sampling times, $b_k$ is the $k$-th element in $\mathbf{b}$, and $z_k^{(i)}$ indicates the $k$-th element in $\mathbf{z}^{(i)}$.*

## 3.3 Understanding EGS

To reveal the serviceability and sampling capability of the developed EGS estimator, according to Definition 1, two basic propositions are given in the following.

**Proposition 1.** *For arbitrary probability vector $\mathbf{p} = [p_1, \cdots, p_K]$ and sampling times $M$, the $K$-dimensional binary code $\mathbf{b} \in \{0,1\}^K$ sampled with EGS always meets*

$$P(b_{k_1} = 1) \le P(b_{k_2} = 1) \Leftrightarrow p_{k_1} \le p_{k_2}, \quad 1 \le k_1, k_2 \le K,$$

*where $P(b_k = 1)$ is the probability of $b_k = 1$, and $P(b_{k_1} = 1) = P(b_{k_2} = 1) \Leftrightarrow p_{k_1} = p_{k_2}$.*

Proposition 1 means that the binary codes sampled with the EGS estimator strictly depend on the probabilities at the corresponding locations. That is, EGS always tends to be a monotonically increasing function in terms of probability, and suffices to act as the binary function $f(\cdot)$.

**Proposition 2.** *For arbitrary probability vector $\mathbf{p} = [p_1, \cdots, p_K]$ and number of sampling times $M$, the EGS estimator is capable of sampling $\binom{K}{M} \times (2^M - 1)$ different binary codes, which includes the whole binary codes with up to $M$ ones and at least 1 one.*

Proposition 2 indicates that the sampling capability of the EGS estimator increases exponentially with $M$. In practice, larger $M$ is always employed to deal with more complex tasks for effect, and smaller one can be utilized to search more lightweight networks for efficiency.

Table 1: Comparison with state-of-the-art image classifiers on CIFAR-10 (lower test error is better).

| Architecture | Test Error (%) | Params (M) | Search Cost (GPU days) | Ops | Search |
|---|---|---|---|---|---|
| DenseNet-BC [22] | 3.46 | 25.6 | - | - | manual |
| PNAS [31] | 3.41 | 3.2 | 225 | 8 | SMBO |
| Hierarchical evolution [33] | 3.75 | 15.7 | 300 | 6 | evolution |
| AmoebaNet-A [44] | 3.34 | 3.2 | 3150 | 19 | evolution |
| AmoebaNet-B + cutout [44] | **2.55** | 2.8 | 3150 | 19 | evolution |
| NASNet-A + cutout [60] | 2.65 | 3.3 | 2000 | 13 | RL |
| ENAS + cutout [42] | 2.89 | 4.6 | **0.5** | 6 | RL |
| DARTS (1-th order) + cutout [34] | 3.00 | 3.3 | 1.5 | 7 | gradient-based |
| DARTS (2-th order) + cutout [34] | **2.76** | 3.3 | 4 | 7 | gradient-based |
| SNAS + mild + cutout [53] | 2.98 | 2.9 | **1.5** | - | gradient-based |
| SNAS + moderate + cutout [53] | 2.85 | 2.8 | **1.5** | - | gradient-based |
| SNAS + aggressive + cutout [53] | 3.10 | 2.3 | **1.5** | - | gradient-based |
| Random search baseline + cutout | 3.29 | 3.2 | 4 | 7 | random |
| DATA ($M = 4$) + cutout | 2.70 | 3.2 | **1** | 7 | gradient-based |
| DATA ($M = 7$) + cutout | **2.59** | 3.4 | **1** | 7 | gradient-based |

Table 2: Comparison with classifiers on ImageNet in the mobile setting (lower test error is better).

| Architecture | Test Error (%) | | Params (M) | FLOPs (M) | Search Cost (GPU days) | Search |
|---|---|---|---|---|---|---|
| | Top 1 | Top 5 | | | | |
| Inception-v1 [48] | 30.2 | 10.1 | 6.6 | 1448 | - | manual |
| MobileNet [20] | 29.4 | 10.5 | 4.2 | 569 | - | manual |
| ShuffleNet-v2 2× [36] | 25.1 | - | ∼5 | 591 | - | manual |
| PNAS [31] | 25.8 | 8.1 | 5.1 | 588 | ∼225 | SMBO |
| AmoebaNet-A [44] | 25.5 | 8.0 | 5.1 | 555 | 3150 | evolution |
| AmoebaNet-B [44] | 26.0 | 8.5 | 5.3 | 555 | 3150 | evolution |
| AmoebaNet-C [44] | **24.3** | **7.6** | 6.4 | 570 | 3150 | evolution |
| NASNet-A [60] | 26.0 | 8.4 | 5.3 | 564 | 2000 | RL |
| NASNet-B [60] | 27.2 | 8.7 | 5.3 | 488 | 2000 | RL |
| NASNet-C [60] | 27.5 | 9.0 | 4.9 | 558 | 2000 | RL |
| DARTS (on CIFAR-10) [34] | 26.7 | 8.7 | 4.7 | 574 | 4 | gradient-based |
| SNAS (mild constraint) [53] | 27.3 | 9.2 | 4.3 | 522 | 1.5 | gradient-based |
| GDAS [18] | 26.0 | 8.5 | 5.3 | 581 | **0.21** | gradient-based |
| DATA ($M = 4$) | 25.5 | 8.3 | 4.9 | 568 | **1** | gradient-based |
| DATA ($M = 7$) | **24.9** | **8.0** | 5.0 | 588 | **1** | gradient-based |

Synthetically, the EGS estimator is capable of improving NAS by tactfully bridging the gap of architectures between searching and validating, while maintaining the differentiability of the NAS pipeline. In practice, the smaller gap guarantees the effectiveness of the inherited architectures from searching to validating, and the differentiability implies that an end-to-end mechanism of searching architectures is achieved for the requisite efficiency.

# 4 Experiments

In this section, we systematically carry out extensive experiments to verify the capability of our model in discovering high-performance architectures. For each task, the experiments consist of two stages, following with the previous work [34, 53]. First, the cell architectures are searched based on our EGS estimator and the best cells are found according to their validation performance. Second, the transferability of the best cells learned on CIFAR-10 [28] and Penn Tree Bank (PTB) [51] are investigated by using them on large datasets, *i.e.*, classification on ImageNet [10] and language modeling on WikiText-2 (WT2) [39], respectively. As a greatly improved work of DARTS, specifically, the experimental settings always inherit from it, except some special settings in each experiment. Specifically, the core code of DATA is released at `https://github.com/XinbangZhang/DATA-NAS`.

## 4.1 Image classification

**Architecture search on CIFAR-10** In our experiments, the candidate primitive set $\mathcal{O}$ includes eight typical operations, *i.e.*, $3 \times 3$ and $5 \times 5$ separable convolutions, $3 \times 3$ and $5 \times 5$ dilated separable convolutions, $3 \times 3$ max pooling, $3 \times 3$ average pooling, identity, and zero. In order to preserve

Table 3: Comparison with state-of-the-art language models on PTB (lower perplexity is better).

| Architecture | Perplexity | | Params | Search Cost | Ops | Search |
|---|---|---|---|---|---|---|
| | valid | test | (M) | (GPU days) | | |
| Variational RHN [57] | 67.9 | 65.4 | 23 | - | - | manual |
| LSTM [40] | 60.7 | 58.8 | 24 | - | - | manual |
| LSTM + skip connections [38] | 60.9 | 58.3 | 24 | - | - | manual |
| LSTM + 15 softmax experts [54] | 58.1 | 56.0 | 22 | - | - | manual |
| DARTS (first order) [34] | 60.2 | 57.6 | 23 | 0.5 | 4 | gradient-based |
| DARTS (second order) [34] | 58.1 | 55.7 | 23 | 1 | 4 | gradient-based |
| ENAS [42] | 68.3 | 63.1 | 24 | 0.5 | 4 | RL |
| Random search baseline | 61.8 | 59.4 | 23 | 2 | 4 | random |
| DATA ($M = 4$) | 58.3 | 56.2 | 23 | 0.5 | 4 | gradient-based |
| DATA ($M = 7$) | **57.1** | **55.3** | 23 | 0.5 | 4 | gradient-based |

their spatial resolution, all operations are of stride one, and the convolutional feature maps are padded if necessary. In EGS, the sampling time $M$ is set to 4 and 7 for a rich search space. During searching, the ReLU-Conv-BN order is utilized in the whole convolution operations, and every separable convolution is always applied twice.

The settings of nodes in our convolutional cell are also following the previous work [60, 44, 31, 34]. Specifically, every cell consists of $n = 7$ nodes, among which the output node is defined as the depthwise concatenation of all the intermediate nodes. The larger networks are always built by stacking multiple cells together. In the $k$-th cell, the first and second nodes are set equally to the outputs in the $(k-2)$-th and $(k-1)$-th cells respectively, with $1 \times 1$ convolution as necessary. Furthermore, the reduction cell with the reduce architecture is utilized at the $1/3$ and $2/3$ of the total depth of the network. The rest of cells are the normal cell with the normal architecture. Specifically, the searched cells are reported in the supplementary material.

**Architecture validation on CIFAR-10** To evaluate the searched architecture, a large network of 20 cells is trained from scratch for 600 epochs with batch size 96 and report its performance on the test set. For fair comparison, we set cutout with size 16, path dropout of probability 0.2 and auxiliary towers with weight 0.4 following exiting works [60, 44, 31, 34]. We report the mean of 5 independent runs for our full model, with different initializations.

Table 1 gives the searched architectures and classification results on CIFAR-10, which shows that DATA achieves comparable results with the state-of-the-art with less computation resources. Such a good performance verifies that DATA can effectively and efficiently search worthy architectures for classification. In DATA, furthermore, higher accuracy is yielded when $M = 7$ compared with $M = 4$. This scenario is in accordance with our motivation that more richer search spaces is beneficial for searching more better architectures. Specifically, the search progresses of different models are showed in Figure 3 (a), which shows that DATA requires less training epochs than SNAS (100 vs 150). A possible reason is that only child architectures are sampled and optimized in DATA, which is faster than optimizing the whole network in SNAS.

**Transferability validation on ImageNet** On ImageNet, the mobile setting where the input image size is 224×224 and the number of multiply-add operations of the model is restricted to be under 600M. An architecture of 14 cells is trained for 250 epoches with batch size 128, weight decay $3 \times 10^{-5}$ and poly learning rate scheduler with initial learning rate 0.1. During training, label smoothing [49] and auxiliary loss [29] are used. Other hyperparameters follow [34] and the mean of 5 independent runs is used for validation.

In Table 2, we report the quantitative results on ImageNet. Note that the cell searched on CIFAR-10 can be smoothly employed to deal with the large-scale classification task. Compared with other gradient-based NAS methods, furthermore, greater margins are yielded on ImageNet. A possible reason is that more complex architectures can be searched in DATA because of the larger search space. Consequently, such more complex architectures handle more complex task on ImageNet better.

### 4.2 Language modeling

**Architecture search on PTB** In the language modeling task, DATA is used to search suitable activation function between nodes. Following the setting in [60, 42, 34], five popular functions including sigmoid, tanh, relu, identity, and zero, are considered in the candidate primitive set $\mathcal{O}$. In

Table 4: Comparison with state-of-the-art language models on WT2 (lower perplexity rate is better).

| Architecture | Perplexity | | Params | Search Cost | Search |
| | valid | test | (M) | (GPU days) | |
| --- | --- | --- | --- | --- | --- |
| LSTM + augmented loss [23] | 91.5 | 87.0 | 28 | - | manual |
| LSTM + cache pointer [16] | - | 68.9 | - | - | manual |
| LSTM [40] | 69.1 | 66.0 | 33 | - | manual |
| LSTM + skip connections [38] | 69.1 | 65.9 | 24 | - | manual |
| LSTM + 15 softmax experts [54] | 66.0 | 63.3 | 33 | - | manual |
| DARTS (searched on PTB) [34] | 69.5 | 66.9 | 33 | 1 | gradient-based |
| ENAS (searched on PTB) [42] | 72.4 | 70.4 | 33 | 0.5 | RL |
| DATA ($M = 4$) | 67.3 | 64.6 | 33 | 1 | gradient-based |
| DATA ($M = 7$) | **66.5** | **64.2** | 33 | 1 | gradient-based |

Table 5: Sensitivity to number of sampling times on CIFAR-10 (lower test error is better).

| Sampling Times ($M$) | 1 | 2 | 3 | 4 | 5 | 6 | 7 | 8 | 9 |
| --- | --- | --- | --- | --- | --- | --- | --- | --- | --- |
| Test Error (%) | 2.94 | 2.95 | 2.78 | 2.70 | 2.72 | 2.60 | 2.59 | 2.50 | **2.45** |
| Params (M) | **2.54** | 2.68 | 2.71 | 3.24 | 3.41 | 3.49 | 3.44 | 3.79 | 3.97 |

the recurrent cell, there are $n = 12$ nodes and the sampling time $M$ is set to 4 and 7 in ensemble Gumbel-Softmax for a rich search space. Similar to ENAS [42] and DARTS [34], in cells, the very first intermediate node is obtained by linearly transforming the two input nodes, adding up the results and then passing through the tanh function, and the rest of activation functions are learned with DATA and enhanced with the highway [58]. The batch normalization [24] in each node to prevent gradient explosion in searching, and disable it during validation. In addition, the recurrent network consists of only a single cell, *i.e.*, any repetitive pattern is not assumed in the recurrent architecture.

**Architecture validation on PTB** In this experiment, a single-layer recurrent network consisting of searched cells is trained with 1600 epoches, and batch size 64 using averaged SGD. Both of the embedding and the hidden sizes are set to 850 to ensure our model size is comparable with other baselines. Other hyper-parameters are set following [34]. For a fair comparison, any additional enhancement is not utilized, such as fine-tuned at the end of the optimization.

Table 3 lists the results in this experiment. From the table, we observe that DATA also is in a position to search recurrent architectures effectively. It empirically shows that the back-propagation algorithm can guide DATA to hit a preferable recurrent architecture, while maintaining the requisite efficiency. Similar to the conclusion in the experiments on CIFAR-10, lower perplexity is achieved when the larger $M$ is used, which verifies that a large search space is also valuable for recurrent architectures .

**Transferability validation on WT2** Different from the settings on PTB, on WT2, we apply embdding hidden sizes 700, weight decay $5\times10^{-7}$, and hidden-node variational dropout 0.15. Other hyperparameters remain the same in the experiment on PTB. In Table 4, the results on WT2 indicate that the transferability is also retentive on recurrent architectures. Conclusively, the consistent results in the above experiments on ImageNet and WT2 strongly demonstrate the transferability on both convolutional and recurrent architectures.

### 4.3 Ablation study

**Sensitivity to number of sampling times** We perform experiments on CIFAR-10 to analyze the sensitivities to the number of sampling time $M$. Table 5 gives the results in this experiment. From this table, it can be observed that larger $M$ indicates higher performance, while more parameters will be introduced as $M$ increases. This is in accordance with the statement in Proposition 2, *i.e.*, more capable networks might be found with larger $M$ to get higher performance.

**Performance on semantic segmentation** We validate the capability of DATA on a more complex task, *i.e.*, semantic segmentation on the PASCAL VOC-2012. In this experiment, we apply the network structure searched on CIFAR-10 as feature extractor and combine it with the head adopted in Deeplab v3 [7]. Compared with NASNet [60] and DARTS [34], DATA achieves better performances with the larger margins, especially when $M$ is large. The results in Table 6 verify that DATA have more prominent superiority on more complex tasks, not just toy tasks on the tiny datasets, because of a large search space that is proportional to the sampling time $M$.

Table 6: Semantic segmentation on the PASCAL VOC-2012 (higher mIOU is better).

| Architecture | NASNet [60] | DARTS [34] | DATA ($M = 1$) | DATA ($M = 4$) | DATA ($M = 7$) |
|---|---|---|---|---|---|
| mIOU(%) | 73.7 | 73.2 | 73.4 | 74.1 | **75.6** |
| Params (M) | 12.4 | 11.8 | **10.8** | 11.7 | 12.7 |

Table 7: Number of operations on CIFAR-10.

| Model | Error (%) | Params (M) |
|---|---|---|
| DARTS(k=1) | $3.00 \pm 0.14$ | 3.30 |
| DARTS(k=2) | $3.10 \pm 0.12$ | 4.00 |
| DARTS(k=3) | $2.95 \pm 0.13$ | 5.20 |
| SNAS | $2.85 \pm 0.02$ | 2.80 |
| DATA(M=1) | $2.94 \pm 0.09$ | **2.54** |
| DATA(M=4) | $2.70 \pm 0.10$ | 3.24 |
| DATA(M=7) | **$2.59 \pm 0.09$** | 3.44 |

Table 8: Validation error on CIFAR-10.

| Model | Search | Child | Gap |
|---|---|---|---|
| DARTS | 12.33 | 45.34 | 33.01 |
| SNAS | 11.46 | 9.33 | 2.13 |
| DATA (M=7) | **11.08** | **9.21** | **1.87** |

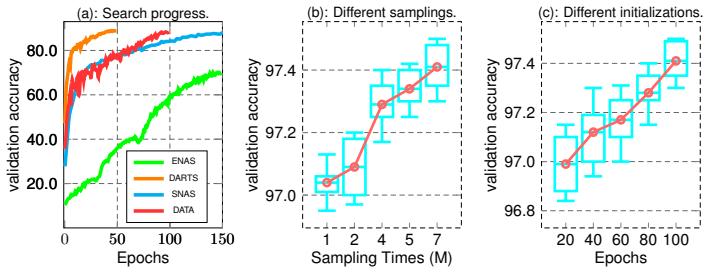

Figure 3: Ablation study.

### Influence of initializations

To investigate the stability of the EGS estimator, we compare the performance of DATA under initializations. Specifically. in Table 7 and Figure 3 (b), the stds of DATA and the variances of DATA with different sampling time (M) are reported, which indicate that DATA (M=1,std=0.08) achieves lower stds than DARTS (k=1,std=0.14). Furthermore, Figure 3 (c) shows the variances of architectures in search process for a comprehensive analysis.

### Contribution of Ensemble Gumbel-Softmax

To explore the contribution of the EGS estimator on DATA, we study the gaps of of the validation errors between search networks and child networks on CIFAR-10. For different models, Table 8 reports the comparison of the validation errors at the end of search and after architecture derivation without fine-tuning. The results show that DATA (gap=1.87, err=9.21) is superior to maintain better performance compared with SNAS (gap=2.13, err=9.33) and especially DARTS (gap=33.03, err=45.34). This demonstrates that the EGS estimator endows DATA the capability of bridging the gap of architectures during searching and validating.

## 5 Conclusion

We present DATA to bridge the gap of architectures during searching and validating in a differentiable manner. For this purpose, the EGS estimator that consists of an ensemble of a group of Gumbel-Softmax estimators is developed, which is in a position to sample an architecture that approaches to the one during searching as close as possible, while guaranteeing the requisite efficiency. By searching with the standard back-propagation, DATA is able to outperform the state-of-the-art architecture search methods on various tasks, with remarkably better efficiency.

Future work may include searching the whole networks with our EGS estimator and injecting the EGS estimator into deep models to handle other machine learning tasks. For the first work, the sampling capability of the EGS estimator guarantees the practicability of searching any networks, but how to further improve the efficiency remains to be solved. For the second work, the differentiability of the EGS estimator indicates that it can be utilized anywhere in networks, *i.e.*, an interesting direction is to recast the clustering process into our ensemble Gumbel-Softmax. By aggregating inputs in each cluster, conclusively, a general pooling for both deep networks and deep graph networks [1, 5, 6] can be developed to deal with Euclidean and non-Euclidean structured data uniformly.

### Acknowledgments

This research was supported by Major Project for New Generation of AI Grant No. 2018AAA0100402, and the National Natural Science Foundation of China under Grants 91646207, 61976208, 61773377, and 61573352. We would like to thank Lele Yu, Jie Gu, Cheng Da, and Yukang Chen for their invaluable contributions in shaping the early stage of this work.

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
