[Reviews · NeurIPS 2019]

Reviewer 1



The paper takes the gumbel softmax trick in SNAS [1] further by ensembling the gumbel softmax estimator. As the result, it has a richer sample space while still being efficient. Rather than the credit assignment approach in SNAS, DATA makes use of the differentiability to update the probability vector. The paper is well written and clearly motivates the proposed approach. I am convinced that the proposed EGS estimator can bridge the gap of architectures between searching and validating, which is a well-known issue in DARTS [2]. The argument that the richer search space of EGS estimator is backed up by the experiments. It clearly outperforms both DARTS and SNAS on image classification and language modelling. However, from my point of view, there is still important analysis I expect to see in the paper. 1. Since the paper makes a claim that the proposed approach bridges the gap of architectures in searching and validating, then I expect to see a table of comparing validation accuracy between search network and child network ( such as Table 1 in [1] ). It will make the claim stronger. The search progress ( Figure. 3 in [1] ) should be attached at least in the supplementary material. 2. It is complained that NAS approach has high variance over different initializations. The results in Table. 1 and Table. 3 are obtained by the mean of 5 independent runs. I don't think it is good enough to just report the mean. A variance plot such as Figure. 3 is expected to see. 3. The interesting part of the proposed of approach is its richer search space. By just looking at the final result in the table, it seems DATA is benefited from the proposed EGS estimator. But it can be further justified by comparing the search result of M=4 and M=7. Unfortunately, I only found the search result of M=1 in the supplementary material. 4. The proposed method is still sampling based, so rather than saying it is differential end to end. I would say the gradient can be obtained by gradient estimator. And the gradient estimator should be written explicitly, at least in the supplementary material. 5. There is no resource constraint in this estimator. SNAS has a regularization on the forward pass time of the child networks [1]. Since the objective of architecture weights and probability vector are both training loss, then is overfitting an issue in this work? One toy example would be if I am trying to fit a polynomial to a regression problem. I want to use NAS method to search for the best polynomial degree. I would imagine searching based on the training loss would lead to largest possible degree which obviously overfits. DARTS in comparison has no such issue because it uses training loss as an objective for the shared weights and validation loss for the architecture vector. 6. It seems DATA should be slower in terms per search iteration than SNAS because it samples multiple times. But DATA is faster than SNAS in the table, can authors provide more intuition on this part? 7. One minor issue is the proposed method is incremental given the existing work [1, 2]. But the work does give the contribution to the NAS community by introducing an efficient estimator which leads to a richer search space. Overall, the paper is an interesting work but leaves some open questions. I don't think it needs further polish and is ready for acceptance yet. After reading the author's response, most of my concerns are addressed. I increase my score to 6. [1]: Xie, S., Zheng, H., Liu, C., & Lin, L. (2019). SNAS: Stochastic Neural Architecture Search. ICLR 2019. [2]: Liu, H., Simonyan, K., & Yang, Y. (2019). DARTS: Differentiable Architecture Search. ICLR 2019.

Reviewer 2



This paper introduces a new approach for Neural Architecture Search by utilizing Gumbel-Softmax operator as a trick to overcome the problem of selecting operators for edges which is discrete and a challenge for gradient descent. The authors justify their proposed approach by saying that the existing approaches do not use the exact architecture that is optimized in validation. Given that each edge might need to use more than a single operator, the authors introduce the idea of ensemble Gumbel-Softmax that allows their framework to be used when multiple operators are required for each edge. The paper is well written, easy to follow and the experimental study shows that their framework performs better than many other state of the arts baselines. One major problem that I have with this work and its motivation is that I am not sure why the operations used in each edge should be mutually exclusive. In general, I guess the gradient descent can determine what combination of operators should be utilized for the best performance. If there is an operator that is not good for an edge, its weight is going to decrease in the optimization. So, not sure what is the practical reason for using one-hot vector after optimization for methods that do so ([33] and [52]). So I believe methods in [33] and [52] can be used without using the one-hot vector step. As mentioned in these papers, they have also used the top-k strongest operations (not only single one). So the justification why the current method is better does is not strong enough even though the authors are showing that their introduced framework performs better and faster.

Reviewer 3



This paper is a natural next step to differentiable architecture search or ENAS like methods. The motivation is to bridge the gap between architectures during search time and validation time. As such, instead of sampling or taking the argmax on each edge, a collection of up to M>1 edges are selected. The results are very good, especially that the proposed method has very low computational cost, while still achieving SOTA or near SOTA results. Especially, if we control for computational efficiency, it achieves superior results over existing baselines. The paper is clearly written and the experiments are on some of the most popular benchmarks.

[Author Response · NeurIPS 2019]

We are very grateful to the constructive comments. Now, we give responses for all questions/comments.

**To reviewer #1:** Thanks for your constructive comments to enhance the completeness of our work.

**Q1:** The comparison of validation error between search network and child network... The search progress...

**R1:** For different models, Table 1 in this response reports the comparison of the validation errors at the end of search
and after architecture derivation without fine-tuning. The results show that DATA (gap=1.87, err=9.21) is superior to
maintain better performance compared with SNAS (gap=2.13, err=9.33) and especially DARTS (gap=33.03, err=45.34),
while requiring less training epochs than SNAS (100 vs 150 epochs), as the search progresses illustrated in Figure 2.

**Q2:** It is complained that NAS approach has high variance over different initializations... A variance plot...

**R2:** In Table 2 and Figure 3, the stds of DATA and the variances of DATA with different sampling time (M) are reported,
which indicate that DATA (M=1,std=0.08) achieves lower stds than DARTS (k=1,std=0.14). Furthermore, Figure 4
shows the variances of architectures in search process for a comprehensive analysis. Please kindly check them.

**Q3:** Search result of M=4 and M=7.

**R3:** The search result (M=4) is shown in Figure 1. More results and analyses will be included in an updated version.

**Q4:** The proposed method is sampling based... And the gradient estimator should be written...

**R4:** Thanks for the suggestion, we will write the gradient estimator in an updated version. In brief, we have:

16
$$\frac{\partial f_{P_k^{(i,j)}}(G^{(i,j)})}{\partial P_k^{(i,j)}} = \frac{\frac{\partial}{\partial P_k^{(i,j)}} \exp\left(\left(\log P_k^{(i,j)} + G_k^{(i,j)}\right)/\tau\right)}{\sum_{l=1}^{K} \exp\left(\left(\log P_l^{(i,j)} + G_l^{(i,j)}\right)/\tau\right)} \left(\delta(k'-k) - \frac{\exp\left(\left(\log P^{(i,j)} + G^{(i,j)}\right)/\tau\right)}{\sum_{l=1}^{K} \exp\left(\left(\log P_l^{(i,j)} + G_l^{(i,j)}\right)/\tau\right)}\right)$$

17
$$= \frac{\delta(k'-k) - f_{P_k^{(i,j)}}(G^{(i,j)})}{\tau P_k^{(i,j)}} f(G_k^{(i,j)}),$$

where $G_k^{(i,j)}$ is the $k$-th Gumbel random variable, $P_k^{(i,j)}$ is the $k$-th element in $\mathbf{P}^{(i,j)} \in \mathbb{R}^K$ and it denotes the probability
of choosing the $k$-th operation on the edge $e^{(i,j)}$. Then, the gradient of the loss with respect to other parameters can be
easily calculated using the chain rule, since the other components in the network are differentiable as well.

**Q5:** Since the objective of architecture weights and probability vector are both training loss... is overfitting an issue...

**R5:** Similar to the dropout to combat overfitting, our EGS provides a network sampling technique that can be treated
as a specific regularization, which is in agreement with the motivation in [C1]. That is, EGS endows DATA with the
capability of searching architectures by learning on the sampled networks. Empirically, our experiments have strongly
evidenced its validity. For a clearer verification, the example you mentioned will be included in an updated version.

**Q6:** DATA should be slower in terms per search iteration than SNAS... can authors provide more intuition on this part.

**R6:** The main reason is that only child architectures are sampled and optimized in DATA, which is faster than optimizing
the whole network in SNAS. As shown in Figure 2, DATA requires less training epochs than SNAS (100 vs 150).

Please kindly check and reassess our paper based on these requested experimental results. Thanks.

Table 1: Validation error on CIFAR-10.

| Model | Search | Child | Gap |
|---|---|---|---|
| DARTS | 12.33 | 45.34 | 33.01 |
| SNAS | 11.46 | 9.33 | 2.13 |
| DATA (M=7) | **11.08** | **9.21** | **1.87** |

Table 2: Number of operations on CIFAR-10.

| Model | Error (%) | Params (M) |
|---|---|---|
| DARTS(k=1) | $3.00 \pm 0.14$ | 3.30 |
| DARTS(k=2) | $3.10 \pm 0.12$ | 4.00 |
| DARTS(k=3) | $2.95 \pm 0.13$ | 5.20 |
| SNAS | $2.85 \pm 0.02$ | 2.80 |
| DATA(M=1) | $2.95 \pm 0.08$ | **2.59** |
| DATA(M=4) | $2.70 \pm 0.10$ | 3.11 |
| DATA(M=7) | $\mathbf{2.59 \pm 0.09}$ | 3.74 |

Figure 1
Normal cell learned on CIFAR-10 (M=4).

Figure 2: Search progress.

Figure 3: Different samplings.

Figure 4: Different initializations.

**To reviewer #2:** (**Q:** Why the operations used in each edge should be mutually exclusive)

**R:** As explained at line125 in the paper, DATA is actually inspired by the fact that operations may NOT be mutually
exclusive but compatible, in contrast to SNAS and DARTS. In SNAS, any probability vector literally converges to an
one-hot vector, which means only one operation is selected. Though DARTS is capable of selecting the top-$k$ strongest
operations, it always subjects to two limitations. First, the number of selected operations $k$ is fixed on each edge, while
DATA can adaptively adjust the numbers on different edges. Second, DATA (gap=1.87) can seamlessly bridge the gap
between architectures in searching and validating, which obviously excels DARTS (gap=33.01), as shown in Table 1.

**To reviewer #2 and #3:** (**Q:** Comparison with a baseline that selects top-$k$ operations during searching)

**R:** The results of selecting top-$k$ operations using DARTS are reported in Table 2, which show that DARTS tends to
select superfluous while inefficient operations that may dramatically introduce more parameters. In contrast, our EGS
endows DATA with the capability of learning simple yet powerful operations, *e.g.*, the skip connection in Figure 1.

**To reviewer #3:** (**Q:** It would be interesting to know what other methods the authors tried... any convergence...)

**R:** Thanks for your interest. We have in the first place explored some naïve methods of sampling multiple operations on
each edge, *i.e.*, selecting top-$k$ operations. However, according to the issues as explained in our response to reviewer #2
and the results in Table 2, none of them works well in practice. For the convergence of DATA, we verify it empirically
in Figure 2 and 4 in this response. It can be seen that our method converges favorably well.

[C1] Bender, Gabriel, *et al.* "Understanding and simplifying one-shot architecture search." ICML. 2018.

[Meta-Review · NeurIPS 2019]

Good paper. Accept. Please update the final version of the paper with the comments and clarifications provided in the rebuttal.